**Subject Area:**
cellular biology

topoisomerase, TOP3B, R-loop, genomic instability

**Authors for correspondence:**
Paul Kalitsis
e-mail: paul.kalitsis@mcri.edu.au
Damien F. Hudson
e-mail: damien.hudson@mcri.edu.au

# Loss of TOP3B leads to increased R-loop formation and genome instability

Tao Zhang[1,2], Mathew Wallis[3,4], Vida Petrovic[5], Jackie Challis[5], Paul Kalitsis[1,2,5] and Damien F. Hudson[1,2]

[1]Murdoch Children's Research Institute, Royal Children's Hospital, Melbourne, Victoria 3052, Australia
[2]Department of Paediatrics, University of Melbourne, Royal Children's Hospital, Melbourne, Victoria 3052, Australia
[3]Tasmanian Clinical Genetics Services, Royal Hobart Hospital, Hobart, Tasmania 7001, Australia
[4]School of Medicine and Menzies Institute for Medical Research, University of Tasmania, Hobart, Tasmania 7001, Australia
[5]Cytogenetics Department, Victorian Clinical Genetics Services, Murdoch Children's Research Institute, Royal Children's Hospital, Parkville, Victoria 3052, Australia

PK, 0000-0001-5569-0609; DFH, 0000-0002-7835-3020

Topoisomerase III beta (TOP3B) is one of the least understood members of the topoisomerase family of proteins and remains enigmatic. Our recent data shed light on the function and relevance of TOP3B to disease. A homozygous deletion for the TOP3B gene was identified in a patient with bilateral renal cancer. Analyses in both patient and modelled human cells show the disruption of TOP3B causes genome instability with a rise in DNA damage and chromosome bridging (mis-segregation). The primary molecular defect underlying this pathology is a significant increase in R-loop formation. Our data show that TOP3B is necessary to prevent the accumulation of excessive R-loops and identify TOP3B as a putative cancer gene, and support recent data showing that R-loops are involved in cancer aetiology.

## 1. Introduction

Topoisomerase III beta (TOP3B) is a member of the IA subfamily of topoisomerases, which unwind negatively supercoiled DNA by cutting a single strand of DNA and passing it through a second single DNA strand [1]. Two TOP3 enzymes exist in higher eukaryotes that have diverged from a single ancestor present in unicellular organisms [2]. TOP3A and TOP3B both unwind negatively supercoiled DNA, but their molecular and *in vivo* functions differ [3]. TOP3A is essential in all species and associates with BLM, RMI1 and RMI2 to form the BTRR complex, which dissolves Holliday junctions that arise during homologous recombination [4]. Unlike TOP3A [5], *Top3b* null mice are viable but present with tissue lesions and a reduced lifespan [6], as well as chromosome instability in spermatocytes and infertility, which becomes more pronounced after successive homozygous breeding [7].

TOP3A and TOP3B have developed distinct specificities for DNA loop (D-loop) and RNA loop (R-loop) structures [8], respectively. In humans, TOP3B possesses dual activities for processing topological problems for both DNA and RNA, whereas TOP3A is DNA specific [9]. D-loops and R-loops are bubble-like structures where one of the strands of DNA (or RNA) is displaced by a homologous strand of DNA (D-loop) or RNA (R-loop) [10,11]. D-loops commonly appear during gene transcription and downstream processing of RNA, DNA repair, DNA replication and meiotic recombination [12]. R-loops are DNA–RNA hybrids that form when nascent RNA hybridizes with the DNA template, leaving the non-template DNA single stranded [13,14]. R-loop formation is highly dependent upon three factors: DNA nicks, high G density and negative supercoiling [15,16]. They most commonly occur when a newly

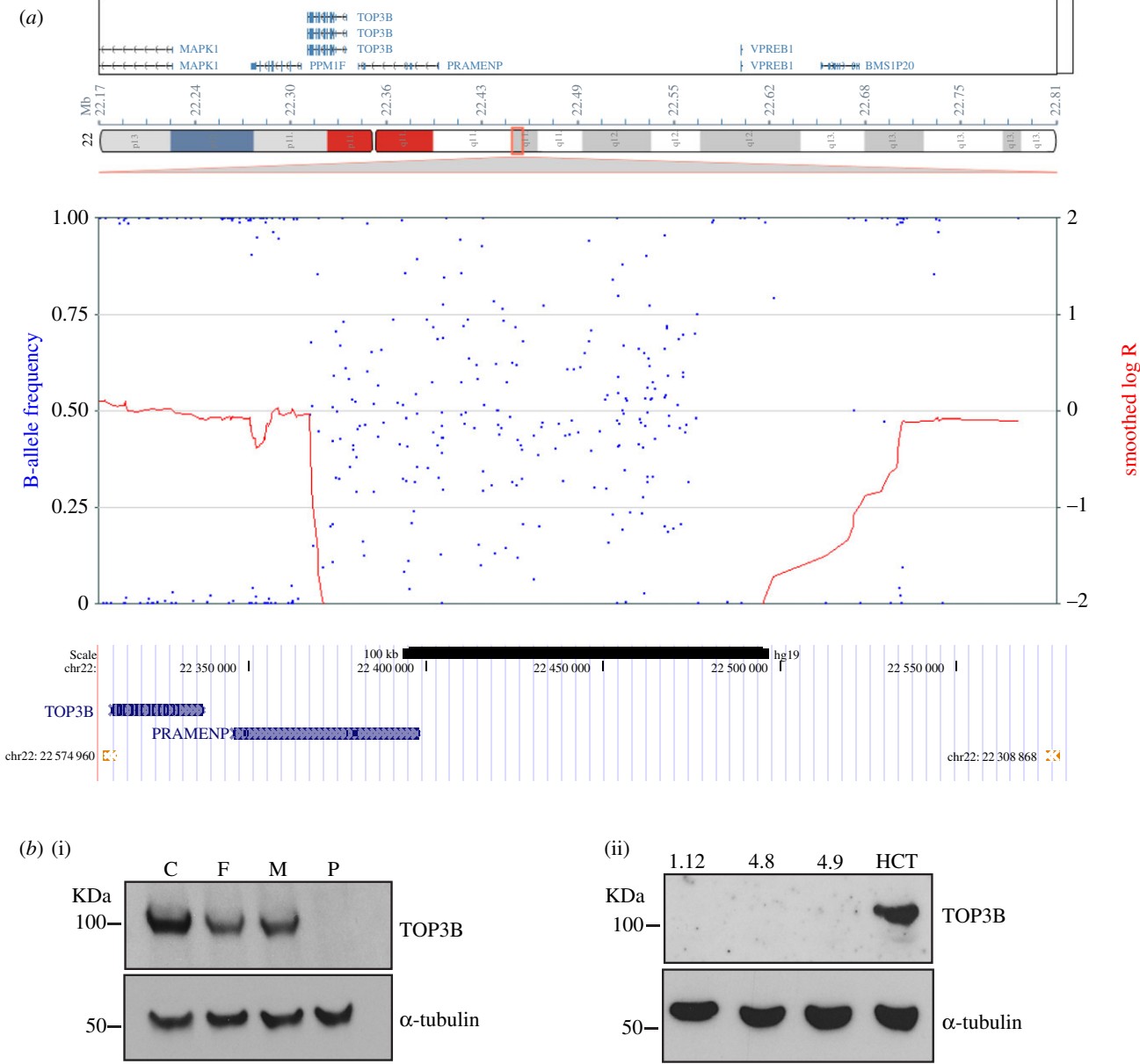

**Figure 1.** Homozygous deletion spanning the TOP3B region. (*a*) Patient genomic DNA was hybridized onto an Infinium Omni2.5 (Illumina) microarray chip for high-resolution copy number analysis. SNP genotyping and copy number levels revealed a homozygous deletion spanning a maximum of 275 kb at chr22: 22 305 007–22 579 821 (human genome build GRCM37/hg19). Note the lack of SNP heterozygosity across the deleted region, shown by blue dots in the B-allele frequency plot. (*b*) (i) Immunoblots of TOP3B and loading control α-tubulin in control (C), parental heterozygote controls (F = father, M = mother) and homozygous TOP3B-deleted patient (P) lymphoblasts. (ii) Immunoblots of TOP3B and α-tubulin in three modelled HCT116 TOP3B null clones (1.12, 4.8 and 4.9), with parental HCT116 (HCT) as a control.

transcribed G-rich RNA anneals back to the C-rich template DNA strand, largely displacing the other DNA strand [17].

It would seem that R-loops would be deleterious to the cell. However, recent data show that R-loops at the 5′ end of genes can protect against DNA methylation, help initiate class switch recombination at the immunoglobulin H (IgH) locus by causing double-stranded breaks in DNA and aid transcription termination at the 3′ end of genes [17–19]. Despite the positive role R-loops play in the cell cycle, excessive R-loop formation can lead to genome instability [16,20].

Several studies implicate TOP3B in processing R-loops. TOP3B binds in a complex with TDRD3, and *Tdrd3* null mice accumulate R-loops at the C-MYC locus [21]. *In vitro* assays in fly and human show that TOP3B activity can relax negatively supercoiled DNA [8,21], an activity that prevents the formation of R-loops.

TOP3B also appears to play an important role in neural cell biology. A study of an isolated northern Finnish family

found that the TOP3B gene was associated with schizophrenia and cognitive impairment [22]. A TOP3B-associated complex has been identified consisting of TOP3B–TDRD3–FMRP (fragile X mental retardation protein) and has been linked to the processing and regulation of neuronal expressed genes [21–23]. The FMRP binds polyribosomal RNA and inhibits the translation of neuronal mRNAs. The TOP3B–TDRD3–FMRP complex prevents the accumulation of transcribed RNAs in cytoplasmic compartments when the cell is stressed [23]. Recent data have further reinforced the pivotal role TOP3B plays in RNA biology. Drosophila S2 cells display defective heterochromatin formation and transcriptional silencing when TOP3B is disrupted, which is reminiscent of mutants in the RNAi-induced silencing complex (RISC) [24].

While new data are emerging for TOP3B, there are still anomalies regarding the protein and its exact functions remain unclear. Here, we present only the second example of a patient without the TOP3B gene. Our new data, using

**Table 1.** TOP3B homozygous deletion frequency was examined in the Catalogue Of Somatic Mutations In Cancer (GRCh38 COSMIC v89, cancer.sanger.ac.uk/cosmic).

| tissues | no. of samples | homozygous deletion | % of samples |
|---|---|---|---|
| adrenal gland | 268 | 2 | 0.746268657 |
| breast | 1544 | 1 | 0.064766839 |
| central nervous system | 1093 | 1 | 0.091491308 |
| endometrium | 598 | 1 | 0.16722408 |
| haematopoietic and lymphoid | 836 | 0 | 0 |
| kidney | 1027 | 1 | 0.097370983 |
| large intestine | 773 | 1 | 0.129366106 |
| liver | 692 | 1 | 0.144508671 |
| lung | 1185 | 2 | 0.168776371 |
| oesophagus | 546 | 1 | 0.183150183 |
| ovary | 729 | 3 | 0.411522634 |
| pleura | 108 | 2 | 1.851851852 |
| skin | 650 | 2 | 0.307692308 |
| soft tissue | 268 | 0 | 0 |
| testis | 152 | 1 | 0.657894737 |
| upper aerodigestive tract | 563 | 0 | 0 |
| total | 11 032 | 19 | 0.172226251 |

further genetic testing was suggested [25]. Indeed, studies in Iceland found that over 60% of patients with renal cancer had a genetic predisposition [26]. The patient had normal *VHL* gene sequencing via the 94 gene Illumina TruSight Cancer Kit on the Illumina MiSeq system, normal sanger sequencing of the *VHL* promoter region, normal *VHL* MLPA (salsa kit P016-C2) testing and normal bioinformatic analysis of other genes known to cause an inherited susceptibility to renal cancer using Illumina TruSight data. Additional testing included normal tumour tissue SDHA/SDHB immunohistochemistry and a normal g-banded karyotype to exclude a translocation involving chromosome 3. SNP-microarray analysis was performed using the high-density Illumina Human Omni2.5 chip. A homozygous deletion was found on 22q11.22 chr22: 22 305 007–22 579 821 (human genome build GRCM37/hg19) that spanned the TOP3B gene (figure 1*a*). The deletion was approximately 275 kb and was driven by 4 kb segmental duplicated regions (99.5% homology) juxtaposed on either side of the deletion (figure 1*a*). Both parents were heterozygote for the deletion. A non-coding RNA with no known function is the only other transcript detected in the deleted area.

To determine the frequency of the TOP3B deletion between the two 4 kb segmental duplicated regions, we searched the region in the gnomAD human structural variant dataset derived from 10 738 genomes from unrelated individuals (gnomad.broadinstitute.org). The allele frequency is 9/21 470 (0.042%), with no homozygous deletions being detected. Interestingly, the allele frequency for duplications is higher at 0.27%, with one homozygous event being identified.

To determine whether homozygous deletions around the TOP3B gene are enriched in cancer mutation datasets, we examined the frequency in the *Catalogue Of Somatic Mutations In Cancer* (GRCh38 COSMIC v89, cancer.sanger.ac.uk/cosmic). We identified 19 with homozygous deletions from 11 032 samples (0.17%) (table 1). TOP3B deletions were found in a broad range of cancer tissue types (table 2). This deletion frequency was markedly elevated compared with the gnomAD dataset frequency of 0.042% for heterozygotes. The increase may reflect the inherent genome instability in cancer genomes, or it is possible that this homozygous variant is a driver in cancer progression. More work needs to be done to investigate this finding.

Two recent case reports, involving similar TOP3B genomic deletions associated with neurological disorders, have been reported [27,28]. Close inspection of the microarray results shows that these structural variants are heterozygous deletions and hence the significance of a single-copy loss remains less significant given the approximate allele frequency of 1/2000 in the general population and up to 1/750 within certain sub-populations. TOP3B genomic deletions are enriched in northeastern sub-isolates compared with the rest of Finland [22]; however, outside of the Finnish population, deletions involving *TOP3B* appear to be seen with approximately equal frequency between cases and controls [29,30].

Immunoblotting with an antibody against TOP3B showed no detectable protein for the homozygous patient (P). Interestingly, heterozygote parents (F and M) displayed a detectable loss of TOP3B protein compared with lymphoblast control cells (C) (figure 1*b*(i); electronic supplementary material, figure S1*a*). To our knowledge, this represents only the second reported case of a homozygous deletion for TOP3B and the first where cell lines have been available for

lymphoblasts from a bilateral renal cancer patient deleted for TOP3B, show multiple hallmarks of genome instability that were pheno-copied in modelled TOP3B null cells of a different lineage. We show TOP3B loss results in excessive R-loop formation, DNA damage and chromosomal instability, and predisposes cells to cancer. Our data also support the growing connection between R-loops and cancer [11,16].

## 2. Results

### 2.1. Identification of a patient with a homozygous deletion of TOP3B

An adult male patient was referred to Adult Cancer Genetics, Austin Hospital, Melbourne with bilateral clear cell renal cancer, diagnosed at the ages of 49 and 52 years. The patient had normal growth, and abdominal and chest CT imaging revealed no cystic lesions of liver or pancreas or lungs. On examination, he was normocephalic with a head circumference of 56.6 cm, and he was not syndromic in appearance. The patient displayed sun-damaged skin with hypopigmented macules and had a number of small (less than 5 mm) hypopigmented pitted scars on his upper chest/back. The proband and his parents did not have schizophrenia or cognitive impairment. He did not have a family history of renal cancer, and his family history was not suggestive of a known renal cancer-predisposing syndrome. However, bilateral renal cancer is rare for a person of 50 years of age, so

**Table 2.** TOP3B homozygous deletions were found in a broad range of cancer tissue types.

| sample name | tissue | chr | start | end | size |
| --- | --- | --- | --- | --- | --- |
| TCGA-OR-A5J2-01 | adrenal gland | 22 | 21 960 940 | 22 199 941 | 239 001 |
| TCGA-OR-A5JD-01 | adrenal gland | 22 | 21 960 940 | 22 219 595 | 258 655 |
| TCGA-A2-A0SW-01 | breast | 22 | 21 953 147 | 22 326 572 | 373 425 |
| TCGA-DB-A64 L-01 | central nervous system | 22 | 21 936 109 | 22 219 595 | 283 486 |
| TCGA-A5-A0G1-01 | endometrium | 22 | 21 960 734 | 22 225 090 | 264 356 |
| TCGA-EU-5905-01 | kidney | 22 | 21 953 147 | 22 225 090 | 271 943 |
| TCGA-AG-A02G-01 | large intestine | 22 | 21 776 250 | 23 147 792 | 1 371 542 |
| TCGA-G3-A25 W-01 | liver | 22 | 21 936 109 | 22 219 595 | 283 486 |
| TCGA-56-8622-01 | lung | 22 | 21 960 717 | 22 225 429 | 264 712 |
| TCGA-63-6202-01 | lung | 22 | 21 960 734 | 22 225 408 | 264 674 |
| OACM5-1 | oesophagus | 22 | 21 833 371 | 23 806 086 | 1 972 715 |
| TCGA-25-1628-01 | ovary | 22 | 21 785 127 | 22 354 061 | 5 68 934 |
| TCGA-61-2018-01 | ovary | 22 | 21 926 720 | 22 187 441 | 260 721 |
| TCGA-61-2109-01 | ovary | 22 | 21 819 244 | 21 967 362 | 148 118 |
| IST-MES1 | pleura | 22 | 21 790 812 | 22 249 007 | 458 195 |
| TCGA-ZN-A9VQ-01 | pleura | 22 | 21 960 940 | 22 200 684 | 239 744 |
| TCGA-GN-A269-01 | skin | 22 | 21 960 940 | 22 225 090 | 264 150 |
| VMRC-MELG | skin | 22 | 21 917 856 | 23 097 569 | 1 179 713 |
| TCGA-2G-AAKL-01 | testis | 22 | 21 953 147 | 22 225 408 | 272 261 |

analyses (we are the second laboratory to report a TOP3B null patient [22]). Sister-chromatid exchange levels were not elevated when compared with parental controls (electronic supplementary material, figure S1*b* left). It is of note that the patient had no history of mental illness. This is in contrast with other studies that have linked the TOP3B gene to autism and schizophrenia [22,23] and juvenile myoclonic epilepsy [28]. A semen sample was taken from the TOP3B null patient and showed a significant reduction in total sperm ($400\,000$ sperm $ml^{-1}$ compared with the normal concentration greater than $15\,000\,000$ sperm $ml^{-1}$) and motility 0% (normal > 50%). The fertility analysis is consistent with a TOP3B null mouse study, which showed a high incidence of aneuploidy in mouse spermatocytes [7].

## 2.2. Loss of TOP3B increases DNA damage and genomic instability

The TOP3B null patient showed no mental illness, but instead presented with bilateral clear cell renal cancer and multifocal and nodular lesions. Genome instability can lead to cancer progression. A common marker of genome instability is heightened levels of micronuclei that usually arise as a result of chromosome loss or fragmented chromosomes not being incorporated into the daughter nuclei. We determined if the loss of TOP3B increased the incidence of micronuclei compared with parental controls (figure 2). Patient lymphoblast null cells, parental heterozygote lymphoblast control cells, CRISPR–Cas9 engineered HCT116 TOP3B null cells and the HCT116 isogenic control cell line were used. HCT116 TOP3B null clones were created independently

(using different guide RNAs) and one null from each guide RNA was used for all subsequent cell biology analyses. HCT116 TOP3B null clones (Top3B_1.12 and Top3B_4.8) were sequenced across the TOP3B Cas9 cut site and no TOP3B protein was detected by immunoblot (figure 1*b*(ii)).

Patient lymphoblast cells significantly increased micronuclei formation compared with parental controls (3.74% in P, 2.03% in M, 1.64% in F and 0.72% in C; figure 2*a*). HCT116 TOP3B null cells showed increased micronuclei formation (figure 2*a*). Overall, the data show that loss of TOP3B causes a mild increase in micronuclei formation, which is suggestive of genome instability. Consistently, null cells from both patient lymphoblast and HCT116 TOP3B$^{-/-}$ cells showed a significant rise in DNA damage, as assessed by immunofluorescence staining in figure 2*b*, using an antibody to γ-H2AX (phosphor S139) that binds specifically at sites of double-stranded DNA breaks [31]. γ-H2AX foci were increased two- to threefold in patient lymphoblast cells and two independent null HCT116 cells compared with controls (figure 2*b*). Terradas *et al*. [32] reported that a high fraction of cells with micronuclei displaying discrete γ-H2AX foci were observed at an early stage post-irradiation. We found a very large rise in the number of cells with micronuclei displaying discrete γ-H2AX foci in HCT116 null cells (electronic supplementary material, figure S1*a*).

Genome instability can arise as a result of defective chromosome segregation. We therefore tested whether null cells showed chromosome bridging defects (figure 3). Bulky DNA bridges are readily visible with conventional DAPI staining while ultrafine DNA bridges (UFBs) are not visible with DNA dyes but can be detected using Plk1-interacting checkpoint helicase (PICH) as a marker [33]. Normal cells

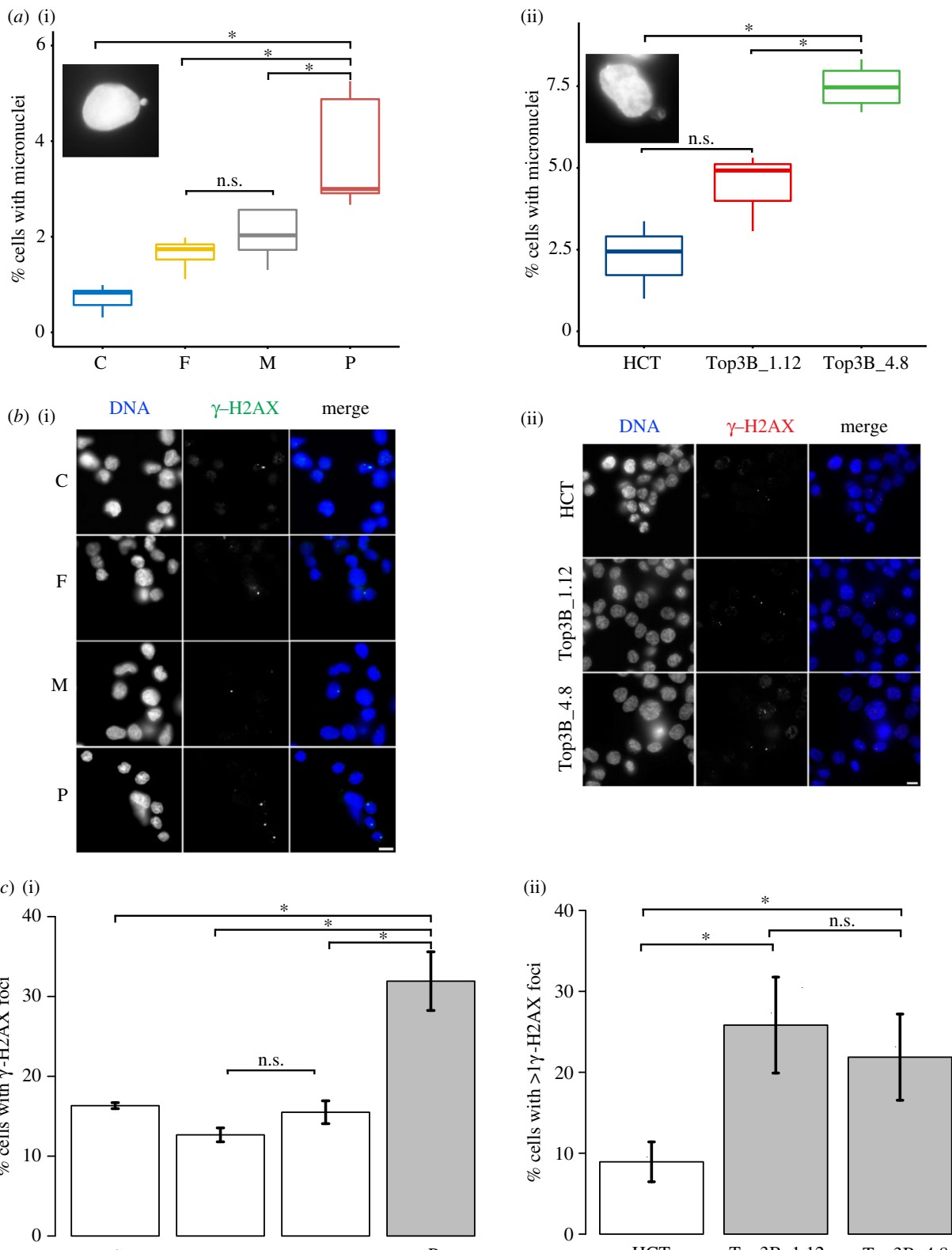

**Figure 2.** Loss of TOP3B results in increased DNA damage. (*a*) Micronuclei levels were determined in (i) control (C), parental heterozygote controls (F = father, M = mother) and homozygous TOP3B-deleted patient (P) lymphoblasts and (ii) wild-type HCT116 (HCT) and TOP3B null clones (Top3B_1.12 and Top3B_4.8). Patient TOP3B null and HCT116 null clones show an increase in micronuclei relative to controls. Inserts are representative images of cells with micronuclei. Data were collected from three independent experiments, with at least 1000 cells scored for each experiment. (*b*) Representative images for (i) lymphoblasts and (ii) HCT116 cells stained for γ-H2AX. (*c*) (i) Patient lymphoblasts and (ii) HCT116 TOP3B null clones show a significant increase in γ-H2AX signal. Data were collected from three independent experiments, with at least 150 cells scored at each experiment. Asterisk denotes $p < 0.05$. n.s., no significant difference. Scale bar, 5 μm.

display a significant amount of UFBs during early anaphase, but these are generally resolved by the BTRR complex and the percentage of cells with UFBs drops markedly in anaphase B [33,34]. Increases in UFBs are often a result of unresolved replication intermediates that persist into mitosis,

while bulky bridges are more commonly formed with problems arising such as under-condensed or entangled mitotic chromosomes during mitosis [34,35]. Our data show a significant rise in anaphase B UFBs in both null patient lymphoblast and HCT116 TOP3B null cells relative to controls

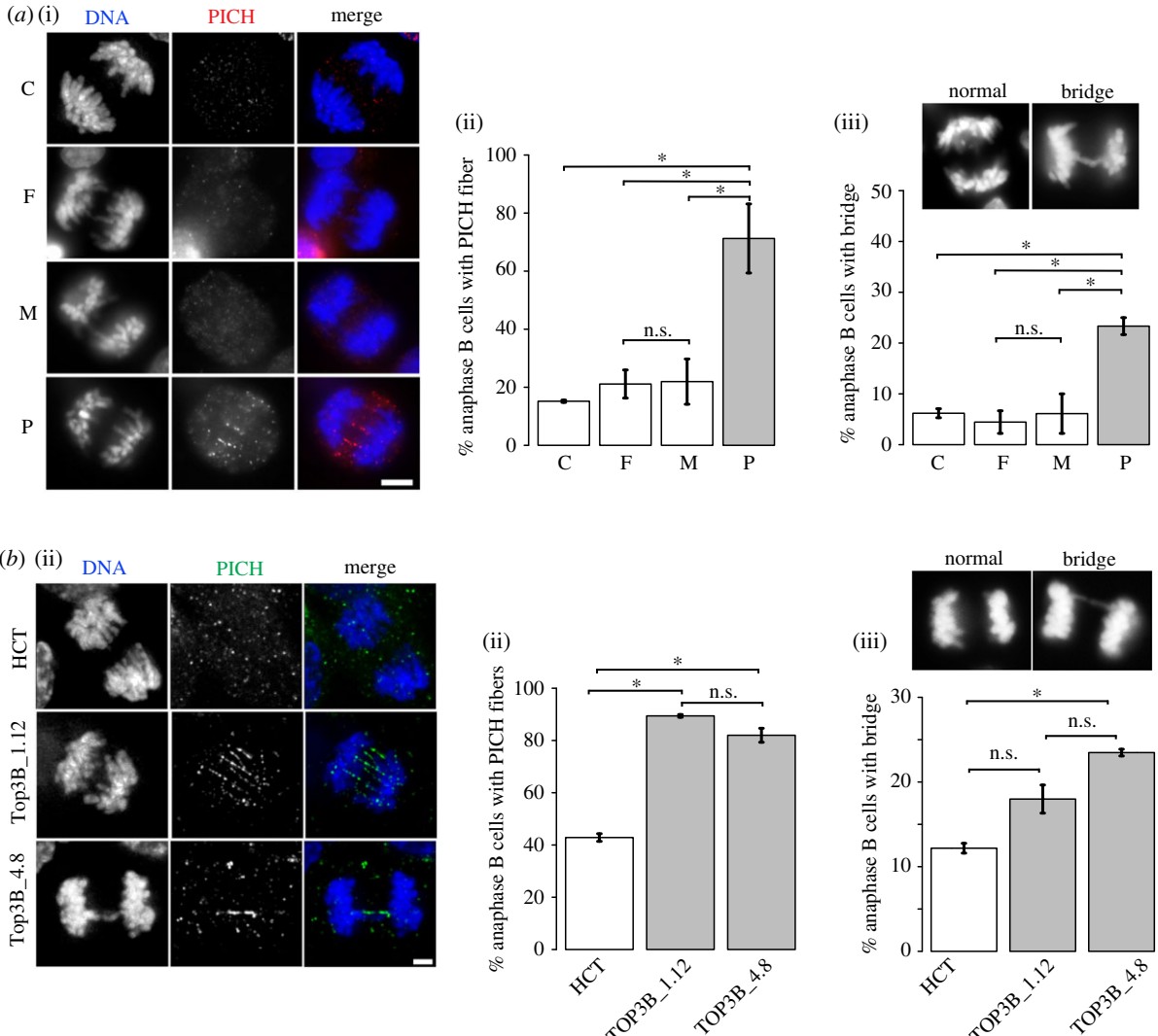

**Figure 3.** Increased DNA bridging as a result of loss of TOP3B. (*a*) DNA bridging was determined by staining lymphoblasts with PICH. (i) Representative images of anaphase B control (C), parental controls (F, M) and patient (P) cells stained with anti-PICH (red) and DAPI for DNA (blue). (ii) PICH bridges were quantitated. (iii) DNA bridging detectable by DAPI in patient and control cells and quantification. (*b*) (i) Representative images of anaphase B TOP3B null HCT116 cells (Top3B_1.12 and Top3B_4.8) stained with anti-PICH (green) and DAPI for DNA (blue). (ii) PICH bridges were quantitated. (iii) DNA bridging detectable by DAPI in control HCT116 (HCT) and HCT116 TOP3B null cells and quantification. Data are from three independent experiments, with at least 50 anaphase B cells scored at each experiment. Asterisk denotes $p < 0.05$. n.s., no significant difference. Scale bar, 5 μm.

(figure 3). Bulky bridges were increased in TOP3B null patient versus control cells (figure 3*a*(iii)), but a statistically significant increase in bulky bridges between null cells and controls in modelled HCT116 cell lines was observed only in one clone (figure 3*b*(iii)).

## 2.3. Loss of TOP3B results in increased R-loops

TOP3B alone has affinity for R-loops and it is also part of the TDRD3 complex, which is involved in processing R-loops. Therefore, we hypothesized that the genome instability we observed was a result of increased R-loop formation. To test this, we used a monoclonal antibody (S9.6) that specifically recognizes RNA : DNA hybrids in a sequence-independent manner [36,37] and analysed patient and HCT116 TOP3B null cells with their respective controls (figure 4). The S9.6 antibody shows especially strong signals at nucleoli, which contain R-loop-prone ribosomal arrays [38]. Treating cells with RNaseH removed the signal, validating the specificity of the S9.6 for the RNA structures (figure 4*a*(i),*b*(i)). The intensity ratio of the S9.6 signal in the nuclei to the DAPI signal,

in both patient lymphoblast null cells and independently targeted HCT116 TOP3B null cells, was significantly higher than their respective controls. The results show that loss of TOP3B causes accumulation of R-loops that probably triggers the observed genomic instability, a consequence that predisposes cells to malignancy [39].

An interesting feature of patient analyses is that the heterozygote parents in certain assays showed an intermediate phenotype between control and null lymphoblasts. This was evident in our micronuclei and R-loop analyses (figures 2*a* and 4*a*). Immunoblotting showed there was a significant reduction of TOP3B protein in parent cells compared with wild-type control cells (figure 1*b*; electronic supplementary material, figure S1*b*), suggesting there could be a level of haploinsufficiency in the heterozygote cells.

## 2.4. TOP3B and the DNA damage response

Replication intermediates arising from incomplete or stalled replication are dealt with by various cellular pathways including non-homologous end joining (NHEJ) [40]. R-loops

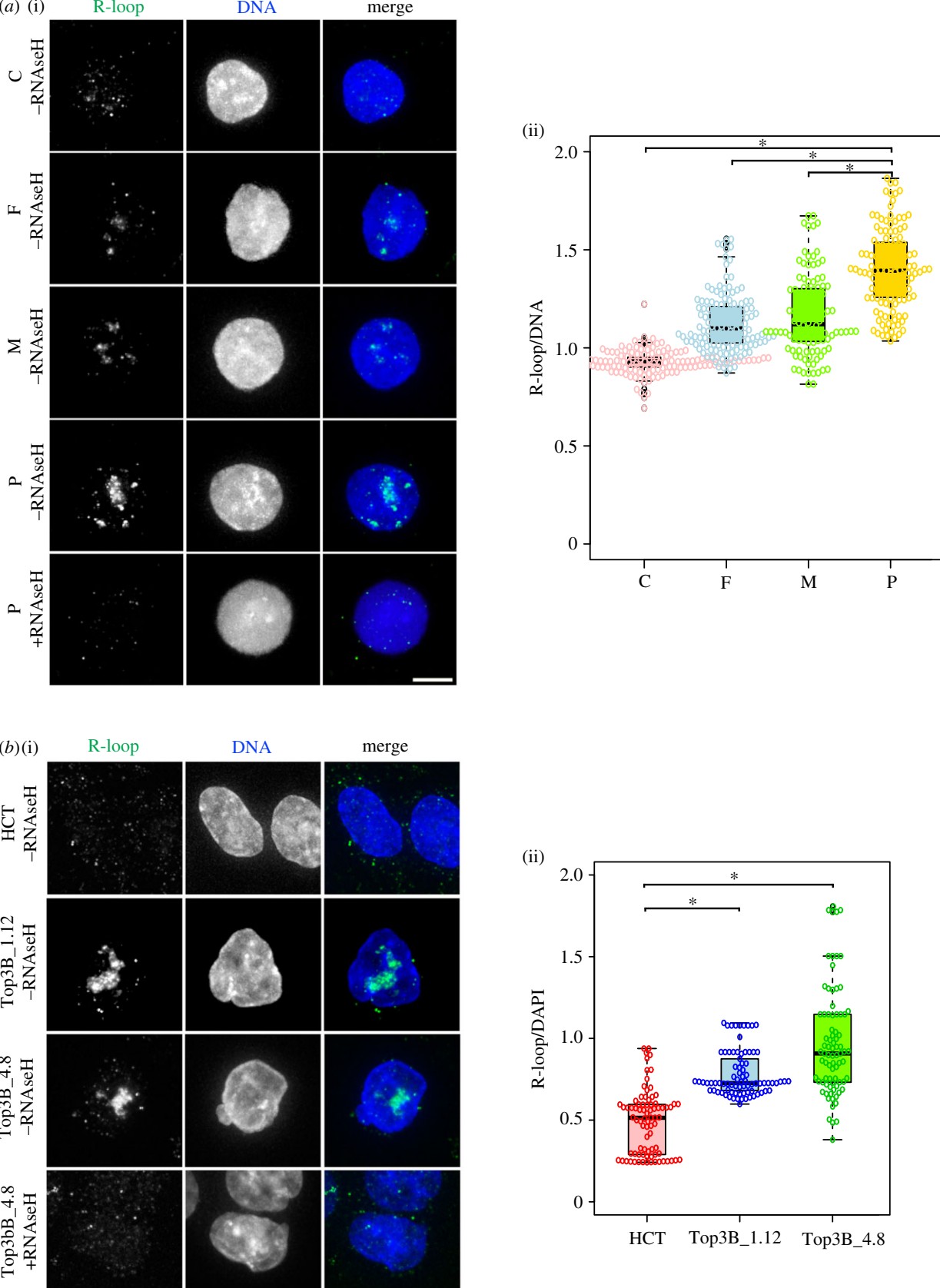

**Figure 4.** Loss of TOP3B causes R-loop accumulation. (*a*) (i) Representative images of lymphoblasts (homozygous control, C; heterozygous parental controls, F, M; homozygous TOP3 null patient, P) and (ii) quantification of R-loop intensity. Significant differences were observed for C versus P, F versus P, M versus P, C versus F and C versus M ($p < 0.05$), but not at F versus M ($p > 0.05$). (*b*) (i) Representative images of wild-type HCT116 (HCT) and homozygous TOP3B null controls (Top3B_1.12 and Top3B_4.8) and (ii) quantification of R-loop intensity. R-loops were stained with the S9.6 antibody (green) and DNA with DAPI (blue). As a control, TOP3B null cells were treated with RNaseH, which disrupts R-loops and causes S9.6 staining to disappear, confirming the specificity of the antibody for R-loops. R-loop intensity was determined as S9.6 intensity relative to DAPI intensity. A significant increase for both the TOP3B null patient (top right) and HCT116 TOP3B null clones (bottom right) was observed. Data are from three independent experiments, with at least 200 cells measured at each experiment. Asterisk denotes $p < 0.05$. Scale bar, 5 μm.

impair replication fork progression and induce replication stress [41]. Flow cytometry profiles showed that TOP3B null cells were not arrested in any particular cell cycle stage (electronic supplementary material, figure S1c). We tested whether TOP3B removal interferes with DNA damage pathways. The p53-binding protein 1 (53BP1) is a key component of the double-strand break signalling and repair pathway in mammals and promotes the pathway for NHEJ-mediated repair of double-stranded breaks (DSBs) [42]. An antibody recognizing 53BP1 was used to stain both HCT116 and HCT116 TOP3B null cells (figure 5a). Interestingly, null cells showed lower levels of 53BP1 than controls, suggesting this pathway is less active with repairing intrinsic replication stress. We then determined whether the repair itself was affected by analysing RAD51, a recombinase that plays a central role in the repair of DSBs via homologous recombination [43,44]. The level of RAD51 in null cells is heightened, consistent with the increase in DNA damage (figure 5b). BRCA1 is another protein involved in the repair of DSBs, but no difference was observed in BRCA1 intensity between null cells and control cells (figure 5c). BRCA1 counteracts 53BP1 by promoting DSB repair using homologous recombination instead of NHEJ [45,46]. Taken together, these results show that elements of the DSB repair signalling pathway are altered when TOP3B is perturbed, and suggest the DNA damage response, as a result of intrinsic replication stress, is diminished when TOP3B is deleted. Also, the homologous recombination pathway may be favoured over NHEJ to repair damaged DNA.

The 53BP1 signalling response in HCT116 TOP3B null cells appears to be lowered. We exposed cells to genotoxic stress and examined the checkpoint responses to further explore this result. HCT116 control and HCT116 TOP3B null cells were treated with replication inhibitors (i.e. thymidine or hydroxyurea) and the damage response was assessed with both γ-H2AX and phospho-P53 (figure 6a,b). Flow cytometry profiles showed that both control and null cells were arrested in G1/S with either thymidine or hydroxyurea treatment (electronic supplementary material, figure S2a). Consistent with earlier experiments, untreated HCT116 TOP3B null cells showed a greater increase in γ-H2AX than controls. However, the situation was reversed with HCT116 TOP3B null cells having a weaker response to DNA damage than controls when treated with thymidine or hydroxyurea (figure 6b(i)). At first glance this seems contradictory, but the response is stronger in wild-type cells than TOP3B null cells when treated with genotoxic agents, suggesting some compromise in the signal response. The higher level of DNA damage in untreated HCT116 TOP3B null cells reflects the fact that under normal circumstances, TOP3B-depleted cells accumulate more DNA damage. Phospho-P53 (Ser15) was used to examine downstream signalling of γ-H2AX. Phospho-P53 at Ser15 is used to amplify the checkpoint response signal. In untreated cells, levels of phospho-P53 were lower in TOP3B null clones than control cells (figure 6b(ii)). This difference was more pronounced in TOP3B null cells than parental control when treated with thymidine or hydroxyurea. Together, these results show that the removal of TOP3B has a deleterious effect on the DNA damage signalling mechanism.

Despite having a perturbed checkpoint response, HCT116 TOP3B null cells still arrest when exposed to genotoxic stress (electronic supplementary material, figure S2a). To assess for recovery after genotoxic stress, cells were treated with thymidine, hydroxyurea or aphidicolin for 24 h, washed and allowed to recover for 7 days. Cells on plates were stained using crystal violet and the number of colony-forming clones was assessed. HCT116 control and HCT116 TOP3B$^{-/-}$ untreated cells showed little difference in colony forming after initially being seeded at equivalent amounts, indicating that the growth rate is not impaired as a result of TOP3B deficiency (figure 7a). This is consistent with our own TOP3B null patient who was of normal height and stature and also mice data that showed nulls develop normally to maturity [6]. However, HCT116 TOP3B null cells formed significantly less clones than controls after being treated with thymidine, hydroxyurea or aphidicolin and allowed to recover for 7 days (figure 7a). This further suggests that the repair mechanism is compromised in TOP3B-deficient cells. We assessed cells for micronuclei formation using the same treatment conditions. Consistent with earlier analyses (figure 2a), there was a slight increase in the amount of micronuclei in untreated HCT116 TOP3B null cells (TOP3B nulls cells versus control: 4.4–7.5% versus 3.5%). However, HCT116 TOP3B null cells treated with thymidine, hydroxyurea or aphidicolin showed a much greater percentage increase (thymidine: 19–24%; hydroxyurea: 17–26%; aphidicolin: 24–27%) in micronuclei relative to controls (thymidine: 10%; hydroxyurea: 11%; aphidicolin: 16%; figure 7b). Consistently, a very large increase in nuclei displaying DNA damage, as assessed by the appearance of γ-H2AX foci, was observed when HCT116 TOP3B null cells were allowed to recover (figure 7c). A large portion of cells retained three to four γ-H2AX foci when recovered from replication stress. Furthermore, more than 60% of cells displayed γ-H2AX foci in the micronuclei (electronic supplementary material, figure S2b). Heterogeneous DNA damage response was observed in cells recovered from replication stress (electronic supplementary material, figure S2c). Together, these results show TOP3B-deficient cells are more sensitive to genotoxic stress and DNA damage.

# 3. Discussion

Our data provide the first direct *in vivo* evidence that TOP3B loss leads to increased R-loop formation. R-loops have gained considerable attention over the last decade and despite there being important benefits of R-loop formation in DNA replication, gene expression, class switch recombination and DNA repair, studies have shown excessive R-loop formation can lead to genome instability [11]. Available data suggest that DSBs can be formed by a collision between R-loops and the replication machinery [47,48]. Our hypothesis is that loss of TOP3B causes increased the formation of R-loops, which results in unresolved recombination intermediates that persist into mitosis and lead to genome instability.

## 3.1. Loss of TOP3B causes genome instability

We found a corresponding increase in genome instability as a result of TOP3B deficiency. This result was consistently identified in our independently targeted HCT116 TOP3B null clones and patient TOP3B null lymphoblasts, strongly confirming our results. Micronuclei were mildly elevated across all cell lines in non-stressed patient TOP3B null lymphoblast cells. This argues for a slow accumulation of DNA damage

royalsocietypublishing.org/journal/rsob Open Biol. 9: 190222

royalsocietypublishing.org/journal/rsob    *Open Biol.* **9**: 190222

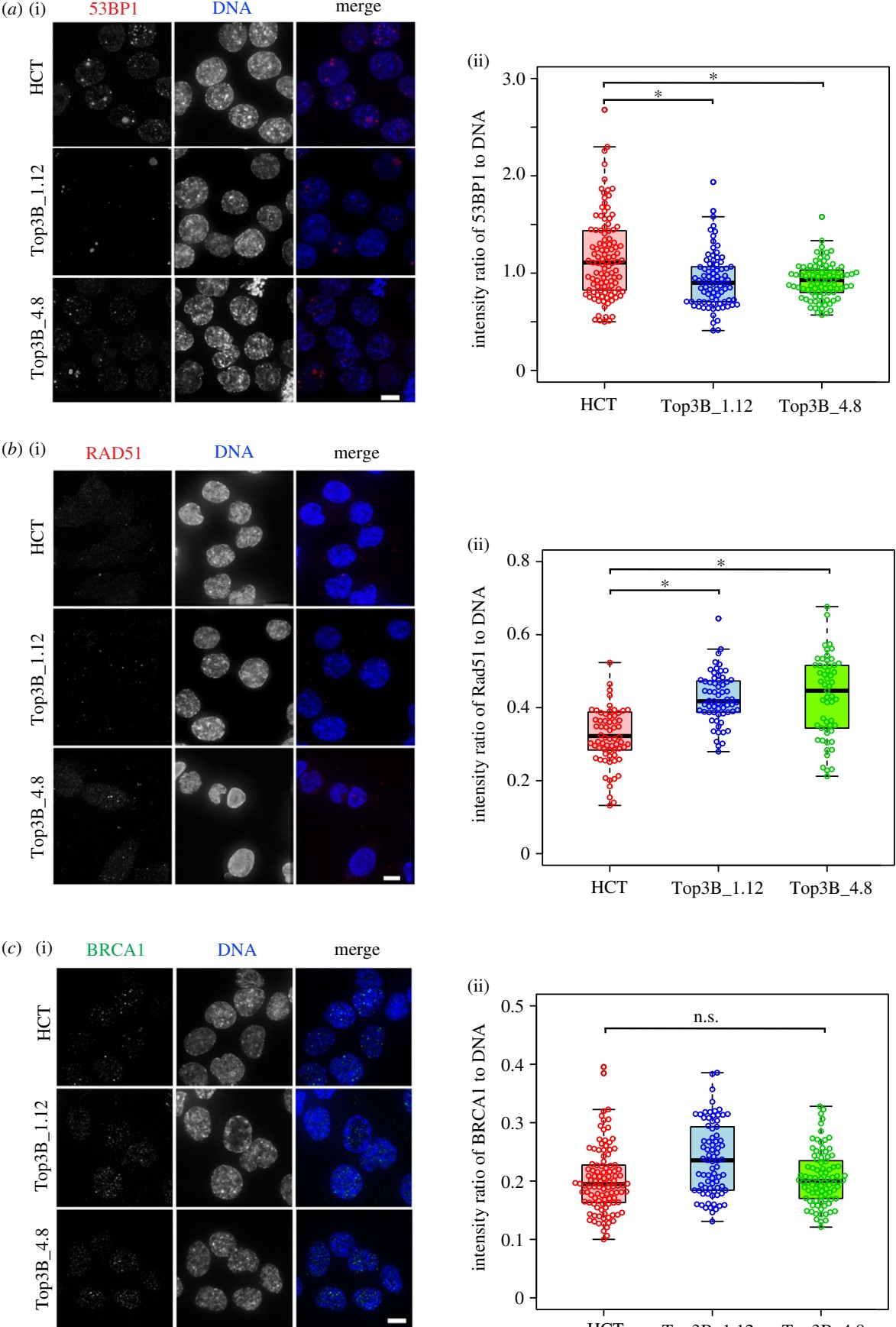

**Figure 5.** Altered DNA damage response in HCT116 TOP3B-deficient cells. (*a*) (i) Representative images of cells stained for 53BP1 (red) and DNA (DAPI, blue). (ii) Quantification reveals a decrease in the intensity ratio of 53BP1 to DAPI in TOP3B null clones (Top3B_1.12 and Top3B_4.8) relative to control HCT116 cells (HCT). (*b*) (i) Representative images of cells stained for RAD51 (red) and DNA (DAPI, blue). (ii) Quantification reveals an increase in the intensity ratio of RAD51 to DAPI in TOP3B null clones relative to control HCT116. (*c*) (i) Representative images of cells stained for BRCA1 (green) and DNA (DAPI, blue). (ii) Quantification in the intensity ratio of BRCA1 to DAPI between control and TOP3B null cells. Data are from three independent experiments, with at least 200 cells measured for each experiment. Asterisk denotes $p < 0.05$. n.s., no significant difference. Scale bar, 5 µm.

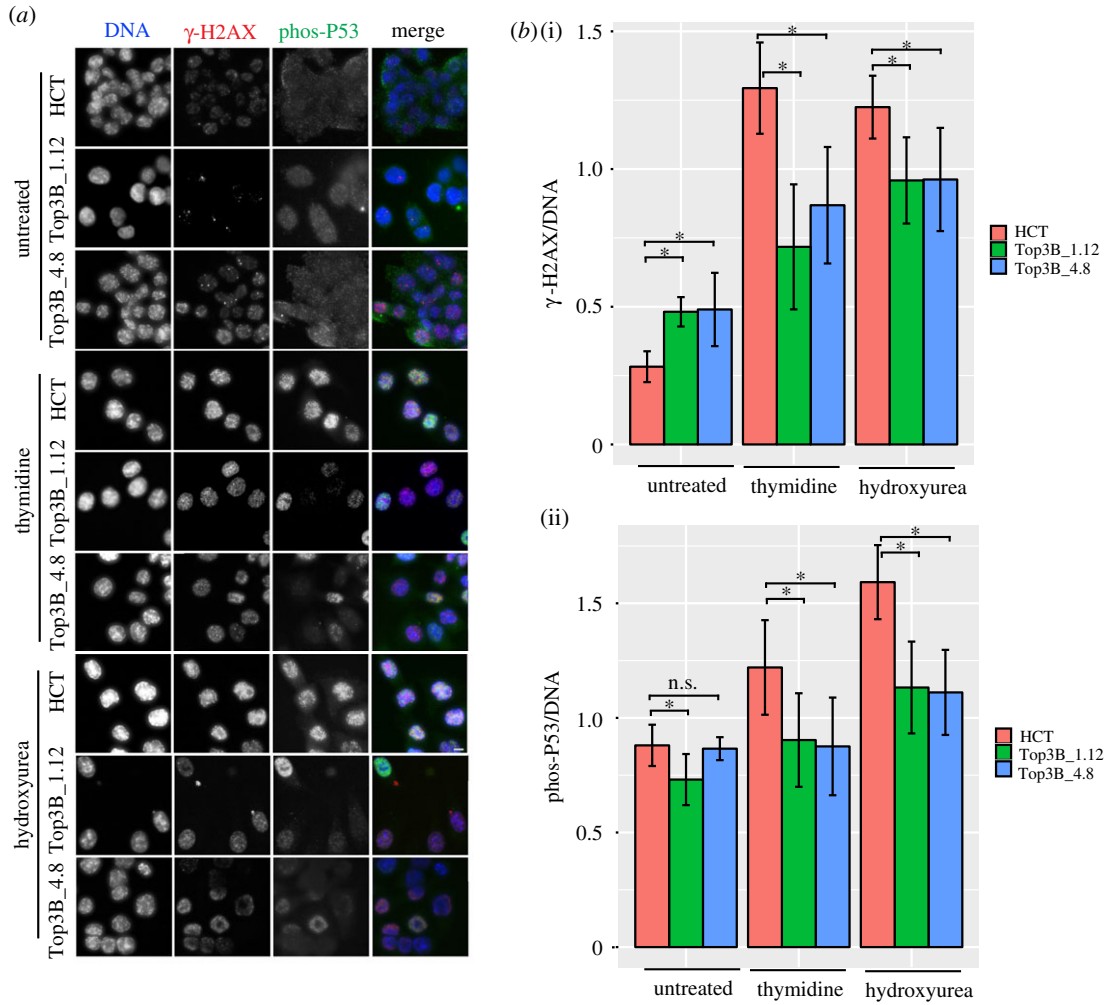

**Figure 6.** DNA damage response of HCT116 TOP3B null cells treated with replication stress agents. (a) Representative images of HCT116 control (HCT) and TOP3B null clones (Top3B_1.12 and Top3B_4.8) untreated and treated with thymidine or hydroxyurea, co-stained for γ-H2AX (red) and phospho-P53 (green). (b) Quantification of the intensity ratio for (i) γ-H2AX to DAPI and (ii) phospho-P53 to DAPI on HCT116 control and HCT116 TOP3B null cells treated with thymidine or hydroxyurea. Data are from three independent experiments, with at least 200 cells measured for each experiment. Asterisk denotes $p < 0.05$. n.s., no significant difference. Scale bar, 5 μm.

over time and suggests that pathological defects such as cancer would be uncommon in less mature subjects. It is useful to contrast this with patients with Bloom syndrome, which similarly also show increased micronuclei and ultrafine bridges [33]. Mutations in the BLM gene lead to patients often being affected by cancers before they reach adulthood [49]. Why does TOP3B disruption not lead to early onset cancers similar to BTRR subunits disruption? Disruption of BTRR subunits and TOP3B causes a significant increase in UFBs and to a lesser extent bulky DNA bridges. However, only disruption of BTRR subunits [34,50–52] causes a dramatic rise in sister-chromatid exchange events, a process that carries an inherent mutability. We would predict those with mutated TOP3B would be likely to develop cancers much later in life, consistent with our patient first diagnosed at 49 years of age.

## 3.2. TOP3B disruption markedly increases R-loop formation and DNA damage

During development, cells are frequently exposed to a variety of environmental stresses. Therefore, cells have evolved elaborate surveillance mechanisms that allow them to transiently halt their progression through the cell cycle, maintain the arrest state and mount a response that should eventually lead to efficient recovery and resumption of the division cycle. Such mechanisms can coordinate the repair of DNA damage, the activation of cell cycle checkpoints to facilitate repair and apoptosis in order to eliminate cells with extensive DNA damage [53]. Thymidine depletes cellular pools of dCTP and causes replication fork stalling [54]; hydroxyurea inhibits the incorporation of nucleotides by interfering with enzyme ribonucleotide reductase and impedes replication [55,56]; aphidicolin interferes with DNA replication by inhibiting DNA polymerases α, ε and δ [57]. These genotoxic agents generate replication stress through different mechanisms, damaging DNA and result in the accumulation of γ-H2AX foci in the cells [58].

Our data show that TOP3B-depleted cells are much more likely to have DNA damage. This was seen in both null patient lymphoblasts and HCT116 TOP3B null cells. DNA damage increases the amount of problems observed in mitosis (e.g. a large rise in ultrafine bridges; figure 3). Ultrafine bridges most frequently arise due to unresolved replication intermediates, which are probably caused by excessive R-loop formation [47,48]. Upon replication stress, TOP3B null cells have a weaker checkpoint response than wide type cells,

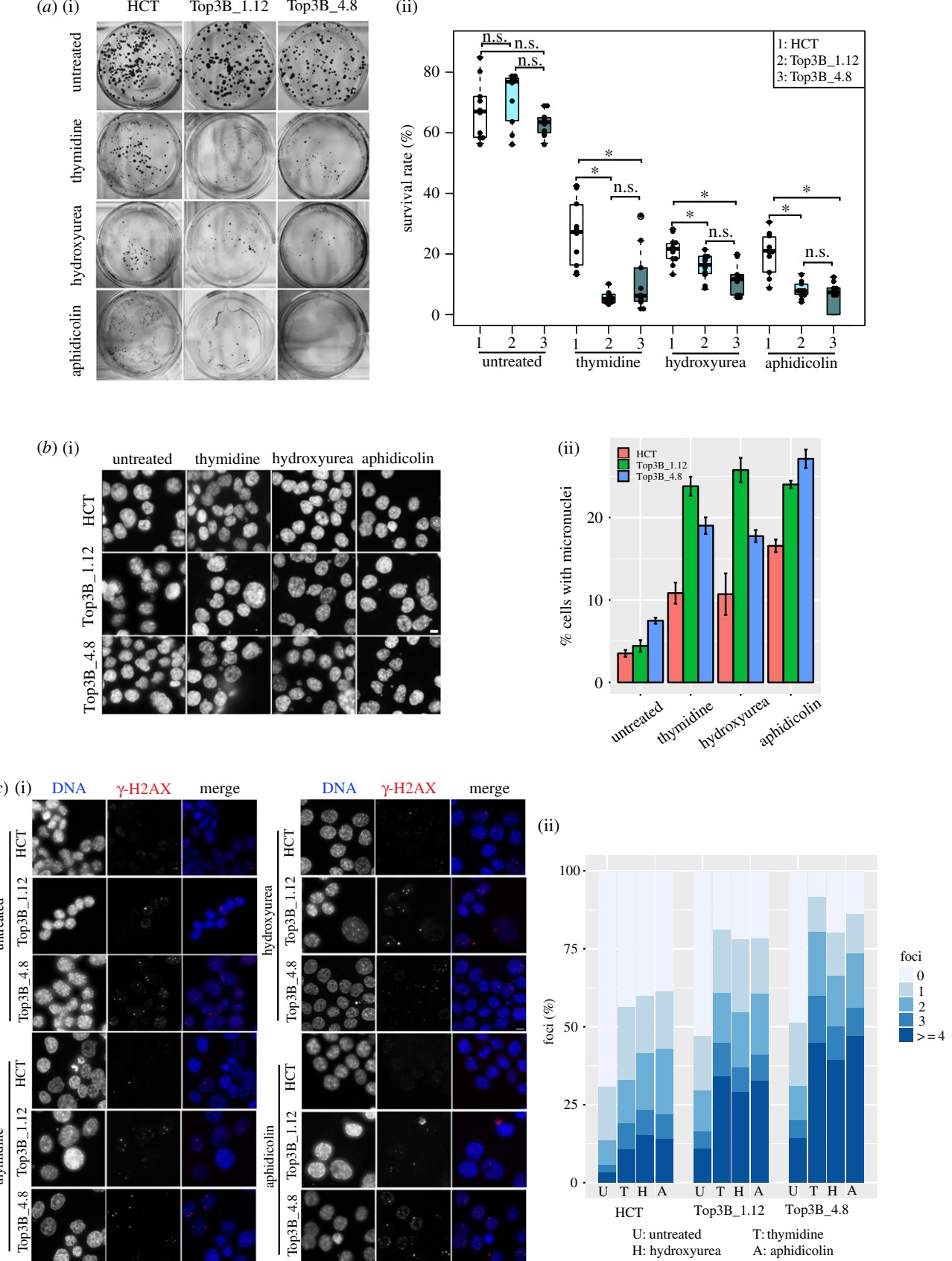

**Figure 7.** Recovery after replication stress is impaired in TOP3B null cells. (*a*) (i) Representative images of crystal violet-stained plates from the colony-forming assay. HCT116 control (HCT) and HCT116 TOP3B null (Top3B_1.12 and Top3B_4.8) cells were treated with no drug, thymidine, hydroxyurea or aphidicolin and allowed to recover for seven days. (ii) Survival rate (%) of colony formation. Data are from three independent experiments, with each experimental sample done in triplicate or more. Asterisk denotes *p* < 0.05. n.s., no significant difference. (*b*) (i) Representative DNA-stained (DAPI) images of micronuclei formation of HCT116 control and HCT116 TOP3B null cells treated with no drug, thymidine, hydroxyurea or aphidicolin and allowed to recover for 48 h. (ii) Quantification of micronuclei. Data are from three independent experiments, with at least 1000 cells scored at each experiment. (*c*) (i) Representative images of HCT116 control and HCT116 TOP3B null clones treated with no drug, thymidine, hydroxyurea or aphidicolin and allowed to recover for 48 h and stained for γ-H2AX (red) and DNA (DAPI, blue). (ii) Quantification of the number of γ-H2AX foci in each cell. Data are from three independent experiments, with at least 150 cells scored at each experiment. Scale bar, 5 μm.

suggesting a compromise in the cell cycle checkpoint control. This is consistent with a previous observation in $Top3b^{-/-}$ murine embryonic fibroblasts [59]. Previous studies have indicated that the formation of micronuclei-staining positive for γ-H2AX is associated with DNA replication stress and irradiation-induced DSBs [32,60]. In this study, we observed increased numbers of cells with micronuclei with discrete γ-H2AX foci (electronic supplementary material, figure S1*a*), further demonstrating the presence of intrinsic replication stress in the TOP3B null cells.

Many proteins involved in DNA damage response and repair have been implicated in the regulation of R-loop accumulation. For example, RAD51 can promote R-loop-mediated DNA damage at R-loops sites, rather than repair DNA damage, leading to genome instability [61]. Patient and engineered TOP3B null cells both had a significant increase in R-loops (figure 4), and our recovery experiments showed a greater increase in DNA damage in null cells after genotoxic stress (figure 7). R-loops play a seemingly contradictory role in the DNA repair pathway. R-loops may promote DSB repair by facilitating homologous recombination and/or NHEJ [62,63]. However, R-loops can sequester BRCA1 binding to DSBs, thereby preventing DNA repair [64,65]. In this study, 53BP1 (NHEJ pathway) levels decreased and RAD51 (homologous recombination pathway) levels significantly increased in HCT116 TOP3B null cells, while levels of BRCA1 foci formation remained unchanged. BRCA1 is a main factor in homologous recombination repair in G2/S, but evidence also indicate that it is involved in NHEJ in G1 [66]. Our results suggest the homologous recombination pathway is activated in response to DSBs in HCT116 TOP3B null cells and preferred over NHEJ as expected in G2/M. A further interesting result is the lower level of phospho-P53 in TOP3B null cells relative to controls, particularly in stressed cells (figure 6). In cells recovered from replication stress, a heterogeneous DNA damage response was observed, with phosphoP53 found to be dislocated from micronuclei displaying γ-H2AX foci (electronic supplementary material, figure S2*c*), suggesting that the DNA damage response in micronuclei is defective [32]. Taken together, these data indicate that TOP3B deletion might also compromise the DNA repair signalling mechanism. The mechanism of TOP3B in DNA repair is an important task for future investigations.

### 3.3. TOP3B and neural defects

The case for TOP3B playing a role in neurogenesis is strong with studies in fly and mice showing neural defects as a result of TOP3B disruption [23] and human TOP3B disrupted patients are predisposed to autism and schizophrenia [22]. TOP3B is enriched at multiple mRNAs with neural functions related to autism and schizophrenia [23]. Our data show increased baseline levels of γ-H2AX in unstressed cells (figure 2*b*), but a reduced response in TOP3B null cells treated with genotoxic agents (figure 6). This, at first glance, seems paradoxical. Nevertheless, a search of the literature shows this trend is strikingly mirrored in multiple studies with cells exposed to DNA damage from schizophrenia patients [67–70]. However, our TOP3B-deficient patient displays no mental illness. This could be explained by the well-described environmental and multigenic events needed for the onset of these mental conditions.

### 3.4. TOP3B and cancer

R-loop processing is gaining more attention in relation to cancer [11]. Our own bioinformatic analyses show a marked increase in homozygous deletions of TOP3B in a broad range of cancer somatic tissues (tables 1 and 2). The link between TOP3B, R-loops and cancer is telling. R-loop formation can increase DSBs, which can, in turn, lead to chromosome translocations or genome instability. Excessive R-loops can also alter gene expression. Therefore, potential tumour suppressor or DNA repair genes could be negatively affected in cells without TOP3B. This is a very interesting new area and we believe over time TOP3B will be added to the list of causative cancer genes.

# 4. Material and methods

## 4.1. Subjects

Written consent was obtained from family members used in this research study as part of the Austin Health Adult Undiagnosed Diseases Program, according to Austin Health policies. The ethic number is AU RED HREC Reference Number: HREC/18/Austin/41.

## 4.2. Genomic microarray

Purified genomic DNA samples were processed by the Illumina Infinium method and were hybridized onto Infinium Omni2.5 v1.2 SNP-array chip (Illumina). Data were analysed with KARYOSTUDIO v1.4 software (Illumina).

## 4.3. Cell lines

Lymphoblasts were obtained from wild-type TOP3B normal (GM19238, Coriell), heterozygous TOP3B$^{+/-}$ parental (mother and father) and a homozygous null TOP3B patient. The near diploid human colorectal cell line HCT116 was engineered to be homozygous null TOP3 using two separate guide RNAs, thereby producing two independently targeted TOP3B null lines (see below). All cell lines were maintained RPMI medium supplemented with 10% fetal bovine serum and penicillin–streptomycin, as described before [34]. All cells grew in incubators with a humidified atmosphere containing 5% $CO_2$ and 95% air at 37°C.

## 4.4. Replication stress

Cells were treated for 24 h with thymidine (2 mM), hydroxyurea (2 mM) or aphidicolin (0.5 µg ml$^{-1}$) to induce replication stress. For recovery experiments, cells were washed with PBS and incubated in the normal medium for another 48 h after stress induction before being harvested for immunofluorescent staining or grown for seven days for the colony-forming assay.

## 4.5. Immunofluorescence microscopy and image analysis

For immunofluorescence staining, HCT116 cells grown on cover slides or lymphoblast cells pre-seeded on poly-L-lysine-coated slides were fixed with 4% paraformaldehyde (PFA) for 10 min, permeabilized with 0.3% triton-X100

and blocked with 3% BSA in PBS. Cells were stained with rabbit polyclonal anti-γ-H2A.X (phosopho S139) (1 : 800) (Abcam), mouse monoclonal anti-PICH (1 : 200) (Millipore), mouse monoclonal anti-53BP1 (1 : 400) (Millipore), mouse monoclonal anti-phospho-p53 (Ser15) (1 : 400) (CST) or mouse monoclonal anti-Lamin A/C (1 : 200) (Chemicon). Secondary antibodies were donkey anti-rabbit Alexa Fluor 488 (1 : 1000) (Invitrogen) or goat anti-mouse Alex Fluor 594 (1 : 1000) (Invitrogen). R-loop staining was performed as described by Schwab *et al.* [71] with slight modifications. Cells were pre-extracted with 0.5% triton-X100 before fixation with 4% PFA. After permeabilization, cells were incubated with 125 µg ml$^{-1}$ RNase A at 37°C for 2 h. For negative controls, cells were further incubated with 0.05 U µl$^{-1}$ RNase H (BioLabs) at 37°C for 4 h. Cells were then blocked with 5% BSA in PBS at room temperature for 2 h before being stained with mouse monoclonal anti-R-loop (S9.6) antibody (1 : 100) (a gift from Dr Andrew Deans, St Vincent's Institute, Melbourne, Australia) at room temperature for 2 h. Cells were mounted with Vectashield mounting medium containing DAPI (Vetor Laboratories).

Images were acquired with a Delta Vision widefield deconvolution microscope. Forty-eight section (0.2 µm per section) images with 20× or 40× objective lens were taken and processed by SoftWoRx 4.1. Cells were further analysed by IMARIS 8.1.2 for immunofluorescence intensity measurements. For intensity measurement, three independent experiments were conducted and at least 150 cells in each experiment were measured. Arbitrary intensity units of Alex488 or Alex594 to DAPI were plotted.

## 4.6. Immunoblotting

Immunoblotting was conducted as described [72]. In brief, cells were resuspended in RIPA with fresh prepared EDTA-free protease inhibitor (Roche) and incubated on ice for 15 min, followed by sonication. Forty micrograms protein from each sample was subjected to 7.5% SDS–PAGE gels. Primary antibodies were rabbit monoclonal anti-TOP3B (1 : 500) (Abcam) or mouse monoclonal anti-TOP3B (1 : 250) (Santa Cruz Biotechnology) or mouse monoclonal anti-α-tubulin (1 : 1000) (Sigma). Secondary antibodies were anti-rabbit IgG-HRP (1 : 10 000) (Amersham) or anti-mouse IgG-HRP (1 : 10 000) (Amersham). Western Blotting Luminol Reagent (Santa Cruz Biotechnology) was used according to the manufacturer's instructions. Intensity was measured by Fiji distribution of ImageJ.

## 4.7. Flow cytometry

FACS analysis was performed, as described in [72], and analysed using FCS express 6.06.

## 4.8. Colony-forming assay

The colony-forming assay was conducted, as described by Crowley *et al.* [73]. In brief, 300 cells/well were seeded in triplicate on 6-well-plates for each sample. After treatment with thymidine, hydroxyurea or aphidicolin for 24 h, cells were washed and allowed to recover for 7 days. After 7 days, 6-well-plates were placed on ice and wash 2× with ice-cold PBS. Cells were fixed with ice-cold methanol for 10 min and stained with a crystal violet staining solution for 5 min at room temperature. The plates were carefully rinsed with ddH$_2$O and allowed to dry at room temperature overnight before imaging. Colonies were counted by Fiji distribution of IMAGEJ with colony_counter.jar plugin (https://imagej.nih.gov/ij/plugins/colony-counter.html). Survival rate = colony numbers/300.

## 4.9. Statistical analyses

Box plots and bar charts were generated using beeswarm R package (https://cran.rproject.org/web/packages/beeswarm/index.html) or gplots (https://cran.r-project.org/web/packages/gplots/index.html). Statistical analyses were conducted using Student's *t*-test (unpaired). *P*-values less than 0.05 were considered as indicating statistically significant differences. Error bars represent SEM.

## 4.10. CRISPR–Cas9 knockout

Two independent nicking CRISPR–Cas9 guide pairs were designed using the CRISPR design tool at crispr.mit.edu. Both pairs targeted the coding sequence of exon 6 of the TOP3B gene (NM_001282112.2). The following target sites for nicking pair no. 1, Guide A minus (5′–GAGAGCGCCTCGTTGTGGTCAGG-3′), Guide B plus (5′-CAGGAGCTGGACCTGCGAATCGG-3′), and no. 4 Guide A minus (5′-CCACTGAGAGCGCCTCGTTGTGG-3′), Guide B plus (5′–GAATCGGCTGTGCATTCACCAGG-3′). Annealed oligonucleotides were cloned into the pSpCas9n (BB)-2A-GFP (PX461) vector (Addgene plasmid ID: 48140) [74]. HCT116 cells were transfected in 6-well trays with Lipofectamine 3000 (Thermo Fisher Scientific) using the supplier's protocol. Two days after transfection, GFP-positive single cells were sorted by FACS into 96-well trays. Genomic DNA from clones was extracted using standard methods followed by PCR amplification screening across the CRISPR target site using the following oligonucleotides; TB3-mf (5′-GTCACAGCTGGCCACTCC-3′) TB3-mr (5′-GAGGGGGACCAGTAGAGG-3′). PCR products were cloned into pGEM-T Easy (Promega) and Sanger sequenced at the Australian Genome Research Facility to confirm the presence of a knockout mutation. Three clones with knockout alleles at the DNA and protein levels were chosen for functional characterization.

## 4.11. Sister-chromatid exchange assay

Fresh blood cells were incubated for three to four days in RPMI 1640 medium/10% FBS with 20 µg ml$^{-1}$ phytohaemagglutin. BrdU (Sigma-Aldrich) was added to a final concentration of 10 µg ml$^{-1}$ for 30 h followed by 0.1 mg ml$^{-1}$ colcemid (Thermo Fischer Scientific) treatment for 45 min before standard metaphase chromosome harvest. Phosphate buffer pH 6.8 was added to cover the dried slides to a depth of 2 mm. Slides were then placed in a biosafety cabinet and were exposed to UV light at a distance of 30 cm for 45 min. The slides were briefly rinsed in dH$_2$O and added to prewarmed 2×SSC at 65°C for 30 min, followed by another rinse in dH$_2$O and stained in Leishman's stain (Sigma-Aldrich).

Ethics. Written consent was obtained from the family members used in this research study as part of the Austin Health Adult Undiagnosed Diseases Program, according to Austin Health policies. The ethic number is AU RED HREC Reference Number: HREC/18/Austin/41.
Data accessibility. This article does not contain any additional data.

**Authors' contributions.** D.F.H. helped conceive, plan experiments and wrote the manuscript. T.Z. helped conceive and performed all cell biology experiments, data analyses and contributed to the manuscript. P.K. helped conceive experiments and generated CRISPR lines, bioinformatic analyses and contributed to the manuscript. M.W. provided clinical data and contributed to the manuscript. V.P. and J.C. performed the microarray analyses.
**Competing interests.** We declare we have no competing interests.

**Funding.** Funding was provided by the National Health and Medical Research Council (Australia) project grants nos. GNT1127209 (P.K. and D.F.H.) and GNT1145188 (P.K. and D.F.H.) and by the Victorian Government's Operational Infrastructure Support Program.
**Acknowledgements.** We are grateful to Dr Andrew Deans (St Vincent's Institute) for the S9.6 antibody and R-loop immunofluorescence staining protocol and Kathy Butler (VCGS) for performing sister-chromatid exchanges assays.

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
