## [Reviewer comments · Open Biology]

Review History

RSOB-19-0222.R0 (Original submission)

Review form: Reviewer 1

Recommendation

Accept with minor revision (please list in comments)

Do you have any ethical concerns with this paper?

No

Comments to the Author

The paper by Zhang et al studied the genetic defect of a patient with bilateral renal cancer and suggested that the homozygous deletion of TOP3B allele contributed to the tumour formation. This is because the loss of TOP3B causes increased the formation of R-loops, which lead to the impairment of replication fork progression and genome instability. More importantly, the author also found that homozygous deletion of TOP3B is enriched in several cancers, providing a putative link between TOP3B deficiency and cancer aetiology.

Overall, I feel that this paper is well written and is supported by the data. I only have a couple of minor suggestions, which are listed below:

- Figure 4a, was the difference between the control group (C) and the parental heterozygote (F & M) significant? No statistic analysis was included.

- In the discussion, the authors compared patients with homozygous TOP3B depletion and BLM mutation, and commented that "only disruption of BLM causes a dramatic rise in sister chromatid exchange events". Has the author performed sister chromatid exchange assay on the TOP3B patient samples and / or the TOP3B depleted HCT cells to confirm?

Review form: Reviewer 2

Recommendation

Accept with minor revision (please list in comments)

Do you have any ethical concerns with this paper?

No

Comments to the Author

In this manuscript, the authors show that TOP3B is necessary to prevent accumulation of excessive R-loops and identify TOP3B as a putative cancer gene and support recent data that R-loops are involved in cancer etiology. I have the following comments.

1. What is the difference between this study and the previous studies (see references 27 and 28)?
2. Authors mentioned several times that this study represents the second reported case of a homozygous deletion for TOP3B. What is the difference between the current example and the first case?

Decision letter (RSOB-19-0222.R0)

28-Oct-2019

Dear Dr Hudson

We are pleased to inform you that your manuscript RSOB-19-0222 entitled "Loss of TOP3B leads to increased R-loop formation and genome instability" has been accepted by the Editor for publication in Open Biology. The reviewer(s) have recommended publication, but also suggest some minor revisions to your manuscript. Therefore, we invite you to respond to the reviewer(s)' comments and revise your manuscript.

Please submit the revised version of your manuscript within 7 days. If you do not think you will be able to meet this date please let us know immediately and we can extend this deadline for you.

To revise your manuscript, log into <https://mc.manuscriptcentral.com/rsob> and enter your Author Centre, where you will find your manuscript title listed under "Manuscripts with

Decisions." Under "Actions," click on "Create a Revision." Your manuscript number has been appended to denote a revision.

- 1) A text file of the manuscript (doc, txt, rtf or tex), including the references, tables (including captions) and figure captions. Please remove any tracked changes from the text before submission. PDF files are not an accepted format for the "Main Document".
- 2) A separate electronic file of each figure (tiff, EPS or print-quality PDF preferred). The format should be produced directly from original creation package, or original software format. Please note that PowerPoint files are not accepted.
- 3) Electronic supplementary material: this should be contained in a separate file from the main text and meet our ESM criteria (see <http://royalsocietypublishing.org/instructions-authors#question5>). All supplementary materials accompanying an accepted article will be treated as in their final form. They will be published alongside the paper on the journal website and posted on the online figshare repository. Files on figshare will be made available approximately one week before the accompanying article so that the supplementary material can be attributed a unique DOI.

Online supplementary material will also carry the title and description provided during submission, so please ensure these are accurate and informative. Note that the Royal Society will not edit or typeset supplementary material and it will be hosted as provided. Please ensure that the supplementary material includes the paper details (authors, title, journal name, article DOI). Your article DOI will be 10.1098/rsob.2016[last 4 digits of e.g. 10.1098/rsob.20160049].

- 4) A media summary: a short non-technical summary (up to 100 words) of the key findings/importance of your manuscript. Please try to write in simple English, avoid jargon, explain the importance of the topic, outline the main implications and describe why this topic is newsworthy.

Images

Data-Sharing

It is a condition of publication that data supporting your paper are made available. Data should be made available either in the electronic supplementary material or through an appropriate repository. Details of how to access data should be included in your paper. Please see <http://royalsocietypublishing.org/site/authors/policy.xhtml#question6> for more details.

Data accessibility section

Sincerely,
The Open Biology Team
mailto:openbiology@royalsociety.org

Reviewer(s)' Comments to Author:

Referee: 1

Comments to the Author(s)

The paper by Zhang et al studied the genetic defect of a patient with bilateral renal cancer and suggested that the homozygous deletion of TOP3B allele contributed to the tumour formation. This is because the loss of TOP3B causes increased the formation of R-loops, which lead to the impairment of replication fork progression and genome instability. More importantly, the author also found that homozygous deletion of TOP3B is enriched in several cancers, providing a putative link between TOP3B deficiency and cancer aetiology.

Overall, I feel that this paper is well written and is supported by the data. I only have a couple of minor suggestions, which are listed below:

- Figure 4a, was the difference between the control group (C) and the parental heterozygote (F & M) significant? No statistic analysis was included.

- In the discussion, the authors compared patients with homozygous TOP3B depletion and BLM mutation, and commented that "only disruption of BLM causes a dramatic rise in sister chromatid exchange events". Has the author performed sister chromatid exchange assay on the TOP3B patient samples and / or the TOP3B depleted HCT cells to confirm?

Referee: 2

Comments to the Author(s)

In this manuscript, the authors show that TOP3B is necessary to prevent accumulation of excessive R-loops and identify TOP3B as a putative cancer gene and support recent data that R-loops are involved in cancer etiology.

I have the following comments.

1. What is the difference between this study and the previous studies (see references 27 and 28)?
2. Authors mentioned several times that this study represents the second reported case of a homozygous deletion for TOP3B. What is the difference between the current example and the first case?

Author's Response to Decision Letter for (RSOB-19-0222.R0)

See Appendix A.

Decision letter (RSOB-19-0222.R1)

01-Nov-2019

Dear Dr Hudson

We are pleased to inform you that your manuscript entitled "Loss of TOP3B leads to increased R-loop formation and genome instability" has been accepted by the Editor for publication in Open Biology.

Article processing charge

Please note that the article processing charge is immediately payable. A separate email will be sent out shortly to confirm the charge due. The preferred payment method is by credit card; however, other payment options are available.

Sincerely,

The Open Biology Team
mailto:openbiology@royalsociety.org

Appendix A

Open Biology
The Royal Society
6-9 Carlton House Terrace
London SW1Y 5AG

31 October 2019

Sub: Submission revised version of the manuscript RSOB-19-0222

Dear Editorial staff,

We are delighted the manuscript has been accepted. As request, we have modified the manuscript in accordance with the reviewers' comments and the required modifications have been introduced in the main text, figure, and figure legends of the revised manuscript. The modifications we have incorporated in the revised version are detailed below. Our point-by-point response to the reviewers' comments are indicated in blue.

We very much appreciate the reviews and excellent suggestions that we believe have improved the quality of the manuscript.

Best regards,

Damien F Hudson

Murdoch Children's Research Institute
Royal Children's Hospital
Flemington Road, Parkville
Victoria 3052 Australia
damien.hudson @mcri.edu.au
Tel: +61 4 0104 1003

Our response to the reviewers' comments:

Reviewer #1

- Figure 4a, was the difference between the control group (C) and the parental heterozygote (F & M) significant? No statistic analysis was included.

Response: In Figure 4a, student's t tests were performed between each groups. Significant differences were observed in C vs F, C vs M, C vs P, F vs P, M vs P. However no significant difference was observed between F and M. We have added the statistical analysis description of C vs F, C vs M and F vs M in the figure legend. Thanks for the reviewer for pointing it out. The significant difference in micronuclei formation and R-loop analyses we observed in C vs F&M could be due to heterozygosity of the parents. Therefore, in the previous manuscript we have included the following justification.

‘An interesting feature of patient analyses is that the heterozygote parents in certain assays showed an intermediate phenotype between control and null lymphoblasts. This was evident in our micronuclei and R-loop analyses (figures 2a and 4a). Immunoblotting showed there was a significant reduction of TOP3B protein in parent cells compared to wild-type control cells (figure 1b and figure S1b), suggesting there could be a level of haploinsufficiency in the heterozygote cells.’ at line 216-221.

In the current manuscript, we add in the significant difference description in the figure legend as suggested (Line 787-788).

- In the discussion, the authors compared patients with homozygous TOP3B depletion and BLM mutation, and commented that “only disruption of BLM causes a dramatic rise in sister chromatid exchange events”. Has the author performed sister chromatid exchange assay on the TOP3B patient samples and / or the TOP3B depleted HCT cells to confirm?

Response: We performed the sister chromatid exchange assay on the TOP3B patient samples. The results were comparable to age matched and parental controls, which had been mentioned in previous manuscript (line 150). Since the sister chromatid exchange levels were not elevated in the patient, we did not perform the assay in the HCT cells. As suggested, we add sister chromatid exchange data into Figure S1b.

Reviewer #2

1. What is the difference between this study and the previous studies (see references 27 and 28)?

Response: For reference 27, a case report of a 12- year old Caucasian female with cognitive problems and dysmorphic features with a 268 kb deletion including the TOP3B gene. The patient was heterozygous TOP3B deletion. The authors suggested that the deletion of TOP3B plays a significant role in the patient's cognitive impairments and possibly craniofacial features.

For reference 28, the patient was a heterozygous TOP3B deletion inherited from the father. The patient was diagnosed with Juvenile Myoclonic Epilepsy presenting with minor cognitive impairments and mild dysmorphic features. The author hypothesized that the TOP3B gene could be a candidate gene in the aetiology of neurodevelopmental disorders including epilepsy.

We are the second laboratory to report homozygous TOP3B deletion patient. Our patient was diagnosed with bilateral clear cell renal cancer and not presenting with mental illness as observed in other heterozygous TOP3B deletion patients in the literature.

2. Authors mentioned several times that this study represents the second reported case of a homozygous deletion for TOP3B. What is the difference between the current example and the first case?

Response: The first report of a homozygous TOP3B deletion patient was by Stoll et al in 2013 (ref 22). Four patients had homozygous TOP3B deletion; two with cognitive impairment and two were diagnosed schizophrenia. No cell line was established in that study and genome instability was not assayed. Subsequently, two more cases with heterozygous TOP3B deletion including ref 27 and 28 were reported. They were all related to neurodevelopmental disorders. We are the second laboratory to report a homozygous TOP3B deletion. However, our patient does not display mental illness. Instead, the patient was diagnosed with bilateral clear cell renal cancer at the ages of 49 and 52, which led to our study of genome instability and the key finding that R-loops are significantly increased.

We added the following changes to the manuscript in addition to the above suggestions from reviewers:

1. 'accumulated' is deleted in current manuscript at line 60.
2. We add (we are the second lab to report a TOP3B null patient [22]) at line 149.
3. We add (figure 1b left) at line 150.
4. We change RNA to RNAs at line 167.
5. We replaced 'were' with 'was' at line 168.
6. '(figure 3a right panel)' are changed to '(figure 3b right panel)' in current manuscript (line 199).
7. We change BLM to BTRR subunits in line 313 and 315.
8. We add three new reference (reference 50, 51 and 52) for BTRR subunits in line 315.
9. 'TOP3B deficient cells had increased R-loop formation' are deleted at line 338.

10. We add 'and Kathy Butler for performing sister-chromatid exchange assays' in the acknowledgement section at line 505 and 506.

11. 'and the sister chromatid exchange assay' are deleted at line 519.